# Proposal for Trapped-Ion Quantum Memristor

**DOI:** 10.3390/e25081134

**Published:** 2023-07-28

**Authors:** Sergey Stremoukhov, Pavel Forsh, Ksenia Khabarova, Nikolay Kolachevsky

**Affiliations:** 1P.N. Lebedev Physical Institute of the Russian Academy of Science, Leninskiy Prospect, 53, 119991 Moscow, Russia; sustrem@gmail.com (S.S.); phorsh@mail.ru (P.F.); k.khabarova@yandex.ru (K.K.); 2Faculty of Physics, Lomonosov Moscow State University, Leninskie Gory, 1/2, 119991 Moscow, Russia; 3National Research Centre “Kurchatov Institute”, Akademika Kurchatova sq. 1, 123182 Moscow, Russia; 4Russian Quantum Center, Bolshoy Bulvar, 30 Bld. 1, 121205 Moscow, Russia

**Keywords:** quantum memristor, cold ions, partial decoherence

## Abstract

A quantum memristor combines the memristive dynamics with the quantum behavior of the system. We analyze the idea of a quantum memristor based on ultracold ions trapped in a Paul trap. Corresponding input and output memristor signals are the ion electronic levels populations. We show that under certain conditions the output/input dependence is a hysteresis curve similar to classical memristive devices. This behavior becomes possible due to the partial decoherence provided by the feedback loop, which action depends on previous state of the system (memory). The feedback loop also introduces nonlinearity in the system. Ion-based quantum memristor possesses several advantages comparing to other platforms—photonic and superconducting circuits—due to the presence of a large number of electronic levels with different lifetimes as well as strong Coulomb coupling between ions in the trap. The implementation of the proposed ion-based quantum memristor will be a significant contribution to the novel direction of “quantum neural networks”.

## 1. Introduction

The name “memristor” (memory resistor) was introduced in the early 1970s in the original work by L. Chua [1]. The main feature of the memristor is the fact that its resistance depends on the charge passed through it. Therefore, this device retains the memory of past states. Chua’s work remained unnoticed until 2008 when D. Strukov and colleagues [2] reported on an experimental implementation of the memristor, which aroused strong interest in the community. Later works challenged Chua’s conceptual line, questioning the memristor’s status as the fundamental fourth element [3]. Despite the difficulties, D. Strukov’s study undoubtedly attracted the attention of researchers. It was quickly realized that memristors had the potential to revolutionize electronics, allowing powerless storage and logic operations [4], as well as simulating the behavior of neuronal synapses [5,6]. These observations have opened up a wide field of applications in physical neural networks and neuromorphic architectures [7,8,9,10,11,12,13,14]. Moreover, the memory formalism [15] is applicable to a wide range of physical and biological systems allowing to be extended far beyond the electronic realm.

In its most general formulation, a memristive device is determined by some input (*x*) and output (*y*) parameters (in the case of a memristor, these are current and voltage, respectively) and the so-called state parameter s(t), which changes according to a certain law over time [16]. These parameters are related by the equations:(1)y=f(s,x)x,
(2)$$\frac{ds}{dt}=g\left(s,x\right),$$
where *f* and *g* are the functions defining the system dynamics. Such memristive devices include, in particular, memcapacitors and meminductors. The input–output characteristic—the dependence of the output parameter *y* on the input *x*—is usually a hysteresis curve.

Mostly, in all memristors designed and studied so far, the input and output signals (*x* and *y*) are classical. A legitimate question arises about the possibility of creating a memristive device that exhibits a hysteresis curve in the input–output characteristic and at the same time has a truly quantum behavior that allows the manipulation of quantum information. According to the current trend, such a device is referred to as a quantum memristor. A quantum memristor has to demonstrate “memristive behavior” for mathematical expectations of quantum observables and must be able to coherently map a quantum input state to an output state. In this regard, a quantum memristor requires the development of an open quantum system in which a quantum device interacts with the environment, for example, via a measurement process. However, this is always associated with certain level of decoherence. This difficulty can be overcome by designing the interaction with the environment in such a way that it is strong enough to provide “memristive behavior”, but weak enough to sufficiently preserve quantum coherence.

To date, the development of quantum memristors is just beginning and their potential impact on neuromorphic computing is difficult to assess. The concept of a quantum memristor was introduced by Pfeiffer et al. [17]. The motivation was to see if both the advantages of quantum computing and memristive behavior could be exploited simultaneously. The advantages of quantum systems provide certain advantages in computation power, while memristivity is a nonlinear feature that goes beyond purely unitary evolution and allows for the implementation of quantum neuromorphic computing.

Two different physical platforms have been studied so far: superconducting circuits [18] and quantum photonics [19,20,21]. Superconducting circuits use memory effects that naturally occur in Josephson junctions [18]. In particular, such effects were used in the recently proposed design of the classical superconducting memristor [22]. In [18], a quantum memristor based on superconductors used quasiparticle tunneling, the control and measurement of which has recently advanced significantly [23,24].

For a quantum photonics platform, the key element of a quantum memristor is a beam splitter with a tunable reflectivity, which is modified depending on the results of measurements in one of the outgoing beams [19]. This feedback mechanism provides memory and nonlinear behavior, while photonic degrees of freedom implement quantum coherence. A similar implementation of a quantum memristor can be achieved using frequency-entangled optical fields and a frequency mixer which creates superpositions of states operating similarly to a beam splitter [20].

The first experimental implementation of a quantum memristor based on an integrated photonic circuit, which was fully reconfigurable using built-in phase shifters, was demonstrated in [21]. This setup is able to exhibit “memristive behavior” of single-photon states using a measurement circuit and classical feedback. The phase shifter is controlled externally by connecting one of the output waveguides through an optical fiber to the micro-controller, which directly controls the phase shifter. The authors also analyzed the possibility of using quantum memristors to increase performance in machine learning problems. They performed numerical simulations of a neural network using purely classical resources and compared it with a network using only three quantum memristors [21]. Numerical results showed that even with such a limited number of quantum memristors, a significant increase in performance can be achieved; an order of magnitude fewer operations were required to train the network using quantum memristors compared to the classical case. Moreover, a quantum network could classify images with up to 95% fidelity, while a classical analogue would only achieve 71% accuracy. Although it is difficult to compare neural networks with each other, whether classical or quantum, these results point to a possible advantage of using quantum memristors in machine learning.

An interesting proposal for the implementation of quantum memristors was made in [25]. It has been shown that the nonlinear dynamics of a non-Markovian nature for quantum memristors can be obtained from the unitary dynamics of a larger system by tracking some of its degrees of freedom associated with auxiliary subsystems. This means that one can obtain efficient families of non-unitary and non-Markovian dynamics in quantum computers by considering a subset of quantum processor qubits. Using this approach, the authors [25] simulated the dynamics of a quantum memristor on a quantum computer and showed that the memristive effect can be achieved via interaction between qubits.

At the moment, not all possible platforms for quantum memristors have been analyzed and demonstrated. In particular, trapped ions, being one of the most successful physical platforms for quantum computations, have not been considered yet. An ionic quantum memristor may have a number of advantages over other counterparts (photonic and superconducting) due to the presence of a sufficiently large number of ion levels with different lifetimes and transitions with different oscillator strengths. Furthermore, an ion-based memristor benefits from straightforward possibility to use common vibrational modes in the trap for ion–ion entanglement. An additional advantage of the ion-based memristive concept is the fact that it can be realized as a standard for ion-based quantum computing and optical atomic clock techniques [26,27].

In this paper, we propose and elaborate the idea of a quantum memristor based on ultracold ions in a Paul trap.

## 2. Trapped-Ion Quantum Memristor

For an ultracold trapped ion one can choose three electronic levels of interest: the ground state |g〉, the excited long-living state |e〉, and the excited short-living state |a〉. Using various techniques (resonant laser or microwave excitation, Raman excitation), it is possible to initiate Rabi oscillations between levels |g〉 and |e〉. The wave function of such a two-level system after the interaction with resonant to the |g〉−|e〉 transition laser field is given by the superposition
(3)$$|\Psi_{in}\rangle =\alpha |g\rangle +\beta |e\rangle ,$$
where ${\left|\alpha \right|}^{2}+{\left|\beta \right|}^{2}=1$. The excitation probability for |e〉 changes with time *t* as [28]
(4)$${\left|\beta \right|}^{2}=\sin^{2}\left(\frac{\Omega t}{2}\right),$$
where Ω—Rabi frequency for |g〉↔|e〉. The formula (Equation 3) represents the wave function of the two-level system interacting with a resonant laser field.

It is also possible to electromagnetically couple the level |e〉 to a level |a〉 using a resonant field. The transition probability can be controlled by varying the laser pulse duration τ and/or its amplitude. The transition |e〉↔|a〉 probability equals [28]
(5)$${\left|\gamma \right|}^{2}=\sin^{2}\left(\frac{\tilde{\Omega }t}{2}\right)$$
with $\tilde{\Omega }$ being the corresponding Rabi frequency for |e〉↔|a〉. Accordingly, the probabilities ${\left|\gamma \right|}^{2}$ and ${\left|\beta \right|}^{2}$ can be changed from 0 to 1. It must be possible to detect the level |a〉 excitation with some technique (make a projection measurement). Figure 1 schematically shows the level scheme.

If one excites |e〉↔|a〉 transition, the wave function of the system can be written as
(6)$$|\Psi_{out}\rangle =\alpha |g\rangle +\beta \sqrt{1-{\left|\gamma \right|}^{2}}|e\rangle +\beta \gamma |a>.$$

As a result, the wave function $|\Psi_{out}\rangle$ describes the three-level system interacting with two resonant laser fields.

The input signal *x* is defined by the mathematical expectation of the ion being in the state |e〉 before laser pulse 2 which excites the transition |e〉→|a〉,
(7)$$x\equiv {\left|\langle e|\Psi_{in}\rangle \right|}^{2}={\left|\beta \right|}^{2}.$$

As for the output signal *y*, we set the mathematical expectation of the ion being in the state |e〉 after laser pulse 1 exciting the transition |e〉→|a〉,
(8)$$y\equiv {\left|\langle e|\Psi_{out}\rangle \right|}^{2}={\left|\beta \right|}^{2}{\left(1-|\gamma \right|}^{2}).$$

In this case, the dependence of the output signal on the input is given by
(9)$$y=(1-|\gamma {|}^{2})x.$$

As already noted, ${\left|\gamma \right|}^{2}$ can be varied over time by the laser 2 pulse shape, so it plays the role of the state parameter *s*, i.e., $s\equiv {\left|\gamma \right|}^{2}$. Equation (Equation 9) takes a form similar to Equation (Equation 1), which is typical for a classical memristor. It is possible to change the state parameter by selecting the duration of laser pulses and/or their amplitude in an arbitrary manner. For the experimental implementation of a quantum memristor, it is necessary to be able to measure *s* during some time interval $T_{i}$ and to organize feedback to change *s* for the next interval $T_{i+1}$. The state parameter *s* changes over time according to an expression similar to (Equation 2):(10)$$\frac{ds}{dt}=\frac{1}{T}\left(x-c\right),$$
where *c* is an arbitrary constant. The dependence (Equation 10) is not the only possible one, it can be chosen in an arbitrary way. It generally comes from the experimental observable behavior. We chose it as a closely corresponding to the equations that describe changes in resistance of classical memristors. By integrating (Equation 10) we obtain
(11)$$s\left(T\right)=s\left(0\right)+\frac{1}{T}\int_{0}^{T}(x-c)dt,$$
where s(0) is the state parameter value at the initial time. The quantum memristor operation procedure is the following. At the initial time, a certain probability s(0) is set. Parameters of the laser 2 pulse are set correspondiongly. The time interval *T* is chosen, during which the transition |e〉→|a〉 is excited and the population of |a〉 is detected subsequently. The probability of the ion being in the state |a〉 in this case equals
(12)$$P_{a}={\left|\langle a|\Psi_{out}\rangle \right|}^{2}=s\left(0\right)x.$$

Hence, by experimental determination of $P_{a}$ one can calculate the input signal *x* on the first time interval *T*. Let us denote this value as $x_{1}$. Using (Equation 9) for $x_{1}$, one can obtain the output signal $y_{1}$. By substituting $x_{1}$ into (Equation 11) one can determine the new state parameter *s*. For this value of s(T), $x_{2}$ and $y_{2}$ are calculated similarly to $x_{1}$ and $y_{1}$, and then the value of s(2T) is found. After that the procedure is repeated. The result is a dependence of the output signal *y* on the input *x*.

The theoretical input–output characteristics calculated for c=0.5 are shown in Figure 2. It can be seen that with a change of *T*, the form of y(x) also changes and for some values of *T* the input–output characteristic demonstrates memristive behavior.

A necessary condition for the implementation of the scheme described above is that the total registration time of the |a〉 state population $\left(\tau_{reg}\right)$ and the duration of the pulse that excites the transition |e〉→|a〉 ($\tau_{2}$) should be much less than the period of Rabi oscillations between levels |g〉 and |e〉:(13)$$\tau_{2}+\tau_{reg}\ll \frac{2\pi }{\Omega }\equiv T_{in}.$$
Inequality (Equation 13) limits the temporal characteristics of the laser field used in the proposed scheme: the temporal width of the second laser should be much less than both the temporal width of the first laser and the period of Rabi oscillations generated by the first laser.

## 3. Population Dynamics under the Action of Two Laser Fields

To study the influence of laser pulse parameters on the memristive characteristics, we simulated the population dynamics of a three-level system under the action of a sequence of resonant laser pulses. For this, we used the system of equations for the population amplitudes *a* considering an atom interacting with two resonant laser fields as described in [29]. In the absence of spontaneous relaxation, this system of equations allows us to calculate the population dynamics of atomic levels for all types of three-level schemes (Λ—scheme, cascade scheme, etc.):(14)$$\frac{da_{|g\rangle }}{dt}=i\Omega^{*}e^{-{\left(\frac{t-t_{01}}{\tau_{1}}\right)}^{2}}a_{|e\rangle },$$
(15)$$\frac{da_{|e\rangle }}{dt}=i\Omega e^{-{\left(\frac{t-t_{01}}{\tau_{1}}\right)}^{2}}a_{|g\rangle }+i\tilde{\Omega }e^{-{\left(\frac{t-t_{02}}{\tau_{2}}\right)}^{2}}a_{|a\rangle },$$
(16)$$\frac{da_{|a\rangle }}{dt}=i{\tilde{\Omega }}^{*}e^{-{\left(\frac{t-t_{02}}{\tau_{2}}\right)}^{2}}a_{|e\rangle },$$
where $t_{02}-t_{01}$ is a time delay between laser pulses and $\tau_{1,2}$ are laser pulse durations ($t_{01}$ is a center of the pulse of the first laser, $t_{02}$ is a center of the pulse of the second laser). By ${}^{*}$, the complex conjugation is indicated. Calculations were carried out for the Rabi frequencies, the pulse durations and the time delays in non-dimensional units.

Figure 3a shows the dynamics of the level population amplitudes calculated for the following parameter values: $\Omega =0.0755,\;\tilde{\Omega }=0.09,\;\tau_{1}=50,\;\tau_{2}=0.5$ and $t_{02}-t_{01}=20$. Due to the impact of the first pulse, Rabi oscillations of the level populations |g〉, |e〉 are observed. Under the action of the second resonant field pulse, the level |a〉 is excited. With given values of the field parameters and by changing the value $\tilde{\Omega }$ in accordance with Equations (Equation 10) and (Equation 11), we calculated *y* and *x*. As a result, a hysteresis dependence of *y* on *x* was obtained as shown in Figure 3b.

Note that this dependence of y(x) was calculated for a three-level system interacting with two resonant laser fields. It corresponds to one of the dependences which were calculated in accordance with (Equation 9) and (Equation 10) (please see Figure 2). It demonstrates a principal opportunity to find values of y and x for a three-level system in the quantum case. It is important that the key control parameter is the time interval *T* in Equation (Equation 10).

Figure 3c shows the dependence of the |e〉 level population calculated for various time delays between pulses. It can be seen that the level population slightly changes with variation of the specified parameter (red curve in Figure 3c). It is important to note that as the duration of the second pulse increases, the population of the |e〉 level can change significantly with the change of the time delay between pulses (black curve in Figure 3c). This may be an additional possibility of obtaining a hysteresis dependence of *y* on *x*.

## 4. Three-Level Scheme for the Case of ^171^*Yb*^+^ Ion

For the experimental implementation of an ion-based quantum memristor, it is necessary to achieve a deeply cooled ion in a Paul trap. For single-ion manipulation, one can use three-dimensional traps, which are somewhat simpler in manufacturing than linear ones and have good optical access to the ion. There are a number of regular methods of ion cooling. For example, Doppler or sympathetic laser cooling [30,31] allows for the achievement of ion temperatures on the order of several mK or even lower. For cooling down to the ground vibrational state, side-band cooling [32] and EIT cooling [33,34,35,36] are successfully used. The most difficult requirement in the experimental implementation of an ion-based memristor is the condition (Equation 13). This difficulty is associated with the detection of the |a〉 state excitation. The observation of the spontaneous decay of this state on a single ion implies the high-fidelity detection of a single photon, which is extremely difficult. As a rule, the detection level population in an ion is determined by the well-known method of electron shelving [37,38,39]. For this the lifetime of the state |a〉 must be much shorter than the lifetime of the state |e〉 to provide signal accumulation and reliable detection of the state |a〉 population. Typically, if one uses an electric dipole transition with a spectral linewidth of tens of MHz, the electron shelving requires several ms for robust measurement.

Accordingly, the lifetime of the level |e〉 should be much longer. For example, in the case of the ${}^{171}Yb^{+}$ ion, the hyperfine sublevels of the ${}^{2}F_{7/2}$ level can be assigned as |g〉 and |e〉, respectively (Figure 4). The energy difference between these states is 3.6 GHz. The level |g〉 can be initially populated by, e.g., π-pulse from the ground state. To couple |g〉 and |e〉 (magnetic–dipole transition) one can use the radiofrequency pulses at 3.6 GHz (standing in for laser pulse 1 in Figure 1). The only decay channel for the ${}^{2}F_{7/2}\left(F=3\right)$ state is the octupole transition to the ground ${}^{2}S_{1/2}\left(F=0\right)$ state. The rate of spontaneous emission at this transition is almost zero (the lifetime equals 6 years), which allows us to arbitrarily set the Rabi oscillation period $T_{in}$ necessary to satisfy condition (Equation 13). In turn, the ground state ${}^{2}S_{1/2}$ can be chosen for the state |a〉. Its population can be straightforwardly detected by electron shelving. A feature of the proposed scheme is that the level |a〉 has less energy compared to |g〉; this is annoying, taking into account the characteristics of the structure of the ${}^{171}Yb^{+}$ ion to fulfill the condition (Equation 13).

Although in the discussion above we considered two laser fields and electric dipole transitions for simulation, the results remain true for the described case of magnetic dipole and electric octupole transitions in the ytterbium ion.

Experimental realization of the proposed ion-base quantum memristor concept is rather complicated: accurate pulse shaping, laser frequency tuning and high-fidelity state reading, in addition to a high-vacuum system with a radiofrequency trap must be under tight control. Furthermore, manipulating within the octupole transition using a narrow-line laser is required, but it can be achieved with the techniques used for ion-based quantum computing [40] and atomic clocks. As for manipulating theoctupole transition, this was also already demonstrated [41].

The necessary states for the implementation of a quantum memristor can be found in other ions as well.

## 5. Coupled Quantum Memristors

As we mentioned above, in the case of ions there is an advantage due to their strong coupling originating from the Coulomb interaction. It is possible to entangle two or more ions. For trapping of the ion chain, it is convenient to use a linear Paul trap. Laser cooling in all directions of motion results in chain crystallization along the trap axis, where the amplitude of the radiofrequency field vanishes. Due to the Coulomb repulsion between the ions, there is a number of ion oscillation normal frequencies. The lowest frequency corresponds to the common vibrational mode of the center of mass. Let us consider a system of two ions, which we designate as the “first” and the “second”. Similar to the case of a single ion, one can initiate Rabi oscillations with a frequency Ω between ${|g\rangle }_{1}$ and ${|e\rangle }_{1}$ of the first ion by means of resonant radiation at the frequency ω. By setting the detuning Ω corresponding to the vibrational mode frequency, one can address adjacent common vibrational mode ${|g\rangle }_{1}{|0\rangle }_{cm}$ and ${|e\rangle }_{1}{|1\rangle }_{cm}$. Here, the symbol ${i\rangle }_{cm}$ denotes the center-of-mass oscillation mode with *i* quanta. The probability of this transition also can be changed by varying the laser pulse shape. Schematically, the level structure is shown in Figure 5.

The transition ${|0\rangle }_{cm}\to {|1\rangle }_{cm}$ indicates the birth of a phonon. The second ion, if it initially was in the state ${|g\rangle }_{2}{|0\rangle }_{cm}$, will change the state to ${|g\rangle }_{2}{|1\rangle }_{cm}$. An experimental proof of the phonon birth can be performed using electron shelving. Then as an input signal *x*, one can chose the mathematical expectation of the first ion being in the state ${|e\rangle }_{1}{|0\rangle }_{cm}$, and as the output signal, the presence of the second ion in the state ${|g\rangle }_{2}{|1\rangle }_{cm}$. The state parameter s(t) is the transition probability ${|0\rangle }_{cm}\to {|1\rangle }_{cm}$, which can be changed over time by implementing the feedback. The dependences y(x) will have a character similar to the corresponding dependences for the case of implementing a quantum memristor on a single ion.

The difficulties of implementing such a scheme include the need of cooling the ions to the ground vibrational state and controlling of the heating of the ions in the trap. The heating rates can be determined from the dephasing of the Rabi oscillations [42] and should be much lower than the memristor coding procedure. The phonon interaction possesses a number of advantages. First, there is the possibility of encoding information on the second ion, i.e., setting the probability of finding the second ion in the state ${|g\rangle }_{2}{|0\rangle }_{cm}$ by exposing it to an additional laser pulse. Second, one can straightforwardly scale up this system up to several tens of ions.

## 6. Conclusions and Perspectives

We propose a quantum memristor based on ultracold ions in a Paul trap. It is shown that at certain parameters of electromagnetic field pulses coupling ion electronic levels, a hysteresis dependence of the output signal on the input is realized. Specific levels of the ${}^{171}Yb^{+}$ ion are considered, which are suitable for the experimental implementation of a proposed quantum memristor. Two options are considered: (i) based on a single ultracold ion and (ii) on a chain of ultracold ions coupled by the common vibrational mode of the center of mass. At the same time, these cases do not cover all the possibilities of implementing an ionic quantum memristor. In particular, it seems promising to use entangled states, which can lead to a significant increase in the capabilities of a quantum memristor, although the experimental technique for such an implementation becomes much more complicated.

Other possible candidates for the quantum memristor realization could be NV centers or other synthesized atoms with specific energy level state properties (selected three levels with corresponding lifetimes). At the same time, an advantage of the ion-based memristive concept is the fact that it can be realized as a standard for ion-base quantum computing and optical atomic clock techniques [26,27]. 

## Figures and Tables

**Figure 1 entropy-25-01134-f001:**
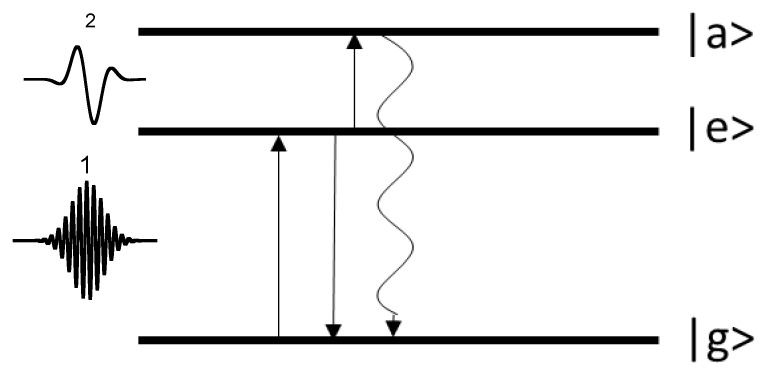
Three-level scheme. Laser radiation 1 excites transition |g〉↔|e〉 with the Rabi frequency Ω. Laser radiation 2 initiates the |e〉→|a〉 transition. The wavy line represents the spontaneous decay of |a〉 state which can be detected.

**Figure 2 entropy-25-01134-f002:**
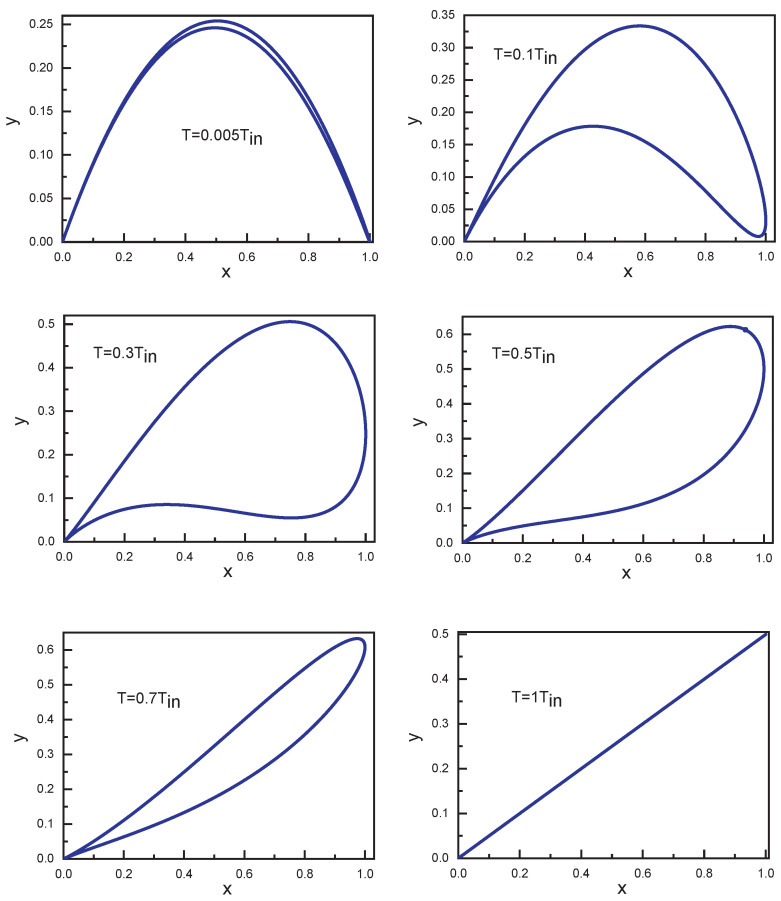
Calculated dependences of *y* on *x* for different durations *T*. Here, $T_{in}=2\pi /\Omega$.

**Figure 3 entropy-25-01134-f003:**
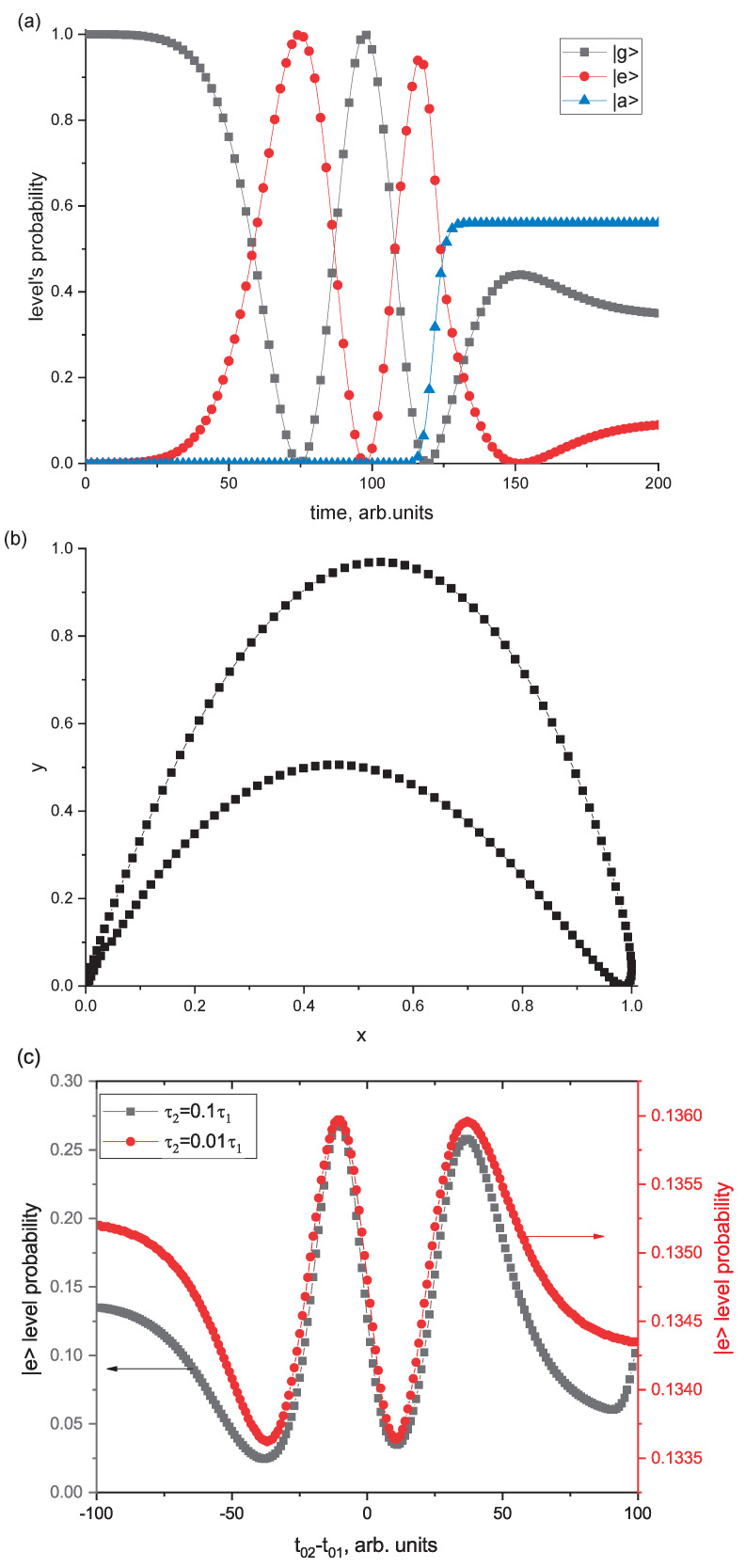
(**a**) Population dynamics of |g〉,|e〉 and |a〉 levels calculated for the following parameters: $\Omega =0.0755,\tilde{\Omega }=0.09,\tau_{1}=50,\tau_{2}=0.5,t_{02}-t_{01}=20$; (**b**) the dependence of *y* on *x* for the Rabi frequency $\tilde{\Omega }$ variation according to (Equation 10) and (Equation 11); (**c**) the dependence of the |e〉 level population for different time delay between pulses, calculated for two values of the second pulse duration (the other values of the parameters are indicated in (**a**)).

**Figure 4 entropy-25-01134-f004:**
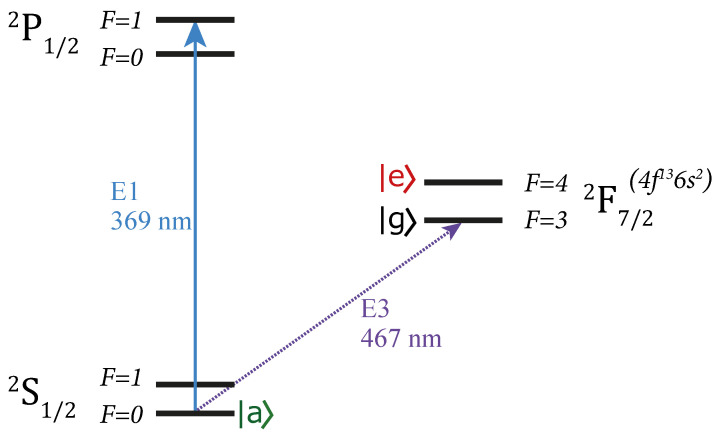
Partial scheme of electronic levels of ${}^{171}Yb^{+}$ ion (not to scale). Hyperfine sublevels of the ${}^{2}F_{7/2}$ state can be assigned as |g〉 and |e〉 states. The level |g〉 can be initially populated by, e.g., π-pulse from the ground ${}^{2}S_{1/2}$ state. The Rabi oscillations between |g〉 and |e〉 can be initiated via radiofrequency pulses at 3.6 GHz. A resonant laser pulse of certain duration couples ${}^{2}F_{7/2}\left(F=3\right)$ state with ${}^{2}S_{1/2}\left(F=0\right)$ state, the latter plays a role of an |a〉 state. The population of |a〉 can be detected by the electron shelving technique at the electric dipole transition at 369 nm.

**Figure 5 entropy-25-01134-f005:**
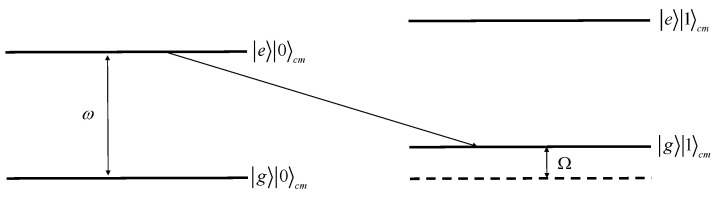
Ion states in the Paul trap, including the lowest vibrational states. ω is the frequency of the transition between levels |g〉 and |e〉; Ω is the frequency of the vibrational mode of the center of mass.

## Data Availability

Raw data can be provided upon request.

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
