# Peer review of "Proposal for Trapped-Ion Quantum Memristor"

_entropy, 2023, doi:10.3390/e25081134_

Round 1
Reviewer 1 Report
The authors present a proposal for the generation of a memristor on a trapped-ion system.
The proposal is of interest but the explanations contains inaccuracies and unclear statements, which makes the present proposal doubtful.
I suggest the authors to strongly revise their proposal.
The inaccuracies and unclear statements can be detailed below.
1) Equation (3) and (4) appear incompatible as defined since beta(tinital)=0 from (4).
I guess the authors want to define (3) during time t from the initial condition alpha(t_i)=1 and define the state as Psi_in. The confusion comes from the fact that Psi_in is is not the initial state but the state after the interaction with laser 1 (during a time t).
This should be reformulated.
2) Similarly psi_out is not clearly defined.
As it appears, it represents the state after interaction with only laser 2, from the initial condition Psi_in.
It should be clearly stated.
3) Equation (9) seems incorrect. It should write y=(1-| gamma|^2)x from (8). The authors should check.
4) Equation (10) is unclear and its development should be made explicit.
Author Response
Reviewer #1
The authors present a proposal for the generation of a memristor on a trapped-ion system. The proposal is of interest but the explanations contains inaccuracies and unclear statements, which makes the present proposal doubtful. I suggest the authors to strongly revise their proposal.
The inaccuracies and unclear statements can be detailed below.
1) Equation (3) and (4) appear incompatible as defined since beta(tinital)=0 from (4).
I guess the authors want to define (3) during time t from the initial condition alpha(t_i)=1 and define the state as Psi_in. The confusion comes from the fact that Psi_in is is not the initial state but the state after the interaction with laser 1 (during a time t).
This should be reformulated.
Reply: We re-formulated the following part of the section 2 of the new version of the manuscript. We believe that now it is more clearly presented.
The following part of the Section 2 was added to the text of the revised manuscript:
“The wave-function of such a two-level system after the interaction with resonant to the |q> -|e> transition laser field is given by the superposition:…
Formulae (3) represent the wave-funtion of two-level system interacted with resonsnt laser field.”
2) Similarly psi_out is not clearly defined.
As it appears, it represents the state after interaction with only laser 2, from the initial condition Psi_in.
It should be clearly stated.
Reply: We re-formulated the following part of the section 2 of the new version of the manuscript. We believe that now it is more clearly presented.
The following part of the Section 2 was added to the text of the revised manuscript:
“As a result, the wave-function |Ψout> describe the tree-level system interacted with two resonant laser fields”.
3) Equation (9) seems incorrect. It should write y=(1-| gamma|^2)x from (8). The authors should check.
Reply: We thank the Reviewer for the comment. It was typo. New version of the manuscript is free from it. We would like to mention, that calculations have been carried out with correct formulae.
4) Equation (10) is unclear and its development should be made explicit.
Reply: We re-formulated the following part of the section 2 of the new version of the manuscript. We believe that now it is more clearly presented.
The following part of the Section 2 was added to the text of the revised manuscript:
“The dependence (10) is not the only possible one, it can be chosen by arbitrary way. It is generally comes from the experimental observable behavior. We choose it as closely corresponds to the equations that describe changes in resistance of classical memristors.”
We thank Reviewer #1 for the useful comments and remarks to the paper. We included the replies to the comments of Reviewer #1 in the revised manuscript, alongside with the replies to Reviewer #2. We hope that, with the above-mentioned modifications of text and figure and the added clarifications, the revised manuscript is now suitable for publication in Entropy.

Reviewer 2 Report
First of all, I would like to say, the article presents an interesting and novel concept for quantum information--- quantum memristors, which could have significant implications for the field. The article is well-organized and easy to follow, with logical connections between sections. From the introduction, we could learn how the concept is developed with its experimental advances. They are clear and concise.
However, for details of their proposal, I have following comments,
1. How is the overhead to implement the “memristors” with this proposal? In my point of view, the technique of non-stop measuring the amplitude of $\ket{a}$ is hard and consuming. If my understanding is right, the proposal requires these measurements and prepared the state with a state parameter form as Eq.(10). As far as I know, they can hardly be implemented with near future techniques. If I am wrong, please let me know.
2. For me, Eq.(6) and Eq.(10) seems the key parts of this protocol. And what we required is a three-level system, where transition between every adjacent energy level can be realized. I think NV center may be a better candidate, with shaped pulse control technology. (But measurement and implementing Eq.(10) are still challenging). So my question is, why is the motivation to claim this proposal a ion-base proposal, as shown in title. Importantly, the part for implantation on Ion trap is too rough. I can hardly get anything on experimental implementation.
3. The simulation is not qualified for this quantum memristors. Fig.2 and Fig.3 are results of simulation of this article. If I am right, Fig.2 presents the results based on Eq.(10) and Eq.(9), and I don’t think it connect to quantum anything or Ion-based platform. Fig.3 presents results of rabi of three level system. For the theme of this article, simulation with ion-based platform on this protocol is required.
Furthermore, there are some explicit typos, for example “seedback” in abstract. And I have question on Eq. (9). As with Eq.(7) and (8) , $y$ should equals to $1-\gama^2 x$.
Overall, the article is well-written and makes a valuable contribution to the field of quantum computing. However, there may be some areas where the authors could provide more detail or clarification. Without that, I cannot support its publication.
Author Response
Reviewer #2
First of all, I would like to say, the article presents an interesting and novel concept for quantum information--- quantum memristors, which could have significant implications for the field. The article is well-organized and easy to follow, with logical connections between sections. From the introduction, we could learn how the concept is developed with its experimental advances. They are clear and concise.
However, for details of their proposal, I have following comments,
- How is the overhead to implement the “memristors” with this proposal? In my point of view, the technique of non-stop measuring the amplitude of $\ket{a}$ is hard and consuming. If my understanding is right, the proposal requires these measurements and prepared the state with a state parameter form as Eq.(10). As far as I know, they can hardly be implemented with near future techniques. If I am wrong, please let me know.
Reply: It is not critical for the ion-base memristor to realize both non-stop lasers action and measuring. Limitations on the temporal characteristics is given by inequality (13).
The following part of the Section 2 was added to the text of the revised manuscript:
“Inequality (13) limits the temporal characterisitics of the laser field used in the propoced scheme: the temporal width of the second laser should be much less than both temporal width of the first laser and period of Rabi oscillations generated by the first laser.”
We also re-organized Figure 1.
- For me, Eq.(6) and Eq.(10) seems the key parts of this protocol. And what we required is a three-level system, where transition between every adjacent energy level can be realized. I think NV center may be a better candidate, with shaped pulse control technology. (But measurement and implementing Eq.(10) are still challenging). So my question is, why is the motivation to claim this proposal a ion-base proposal, as shown in title. Importantly, the part for implantation on Ion trap is too rough. I can hardly get anything on experimental implementation.
Reply: The Reviewer is right that NV center can be one of the base for quantum memristors. We added it at the Conclusions and Perspectives section. We chosen ion-base proposal since it is on the one hand third “standard” for quantum computing platform (along with superconducting Josephson junctions and quantum photonics) and on the other hand there are no proposal for it application to the memory effects (as it for superconducting circuits [18 ] and quantum photonics [19 – 21] as we mentioned in the Section 1).
The following part of the Section 1 was added to the text of the revised manuscript:
“Additional advantage of the ion-base memristive concept is in the fact that it can be realized in standard for ion-base quantum computing and optical atomic clock techniques [26, 27].”
The following part of the Section 4 was added to the text of the revised manuscript:
“Experimental realization of the proposed ion-base quantum memristor concept is really complicated, but it can be done with the technique used for ion-base quantum computing and atomic clocks”.
The following part of the Section 6 was added to the text of the revised manuscript:
“Other possible candidate for the quantum memristor realization could be NV centers or other syntesed atoms having specific energy level state properties (selected three levels with corresponding lifetimes which populations are convenient to drive with laser fields). At the same time an advantage of the ion-base memristive concept is in the fact that it can be realized in standard for ion-base quantum computing and optical atomic clock techniques [26, 27].”
- The simulation is not qualified for this quantum memristors. Fig.2 and Fig.3 are results of simulation of this article. If I am right, Fig.2 presents the results based on Eq.(10) and Eq.(9), and I don’t think it connect to quantum anything or Ion-based platform. Fig.3 presents results of rabi of three level system. For the theme of this article, simulation with ion-based platform on this protocol is required.
Reply: The Reviewer is right to say that not all presented results of simulations are related directly to the quantum memristors. Fig. 2 represent results based on Eq.(10) and Eq.(9). It demonstrates principal opportunity to find values of y and x for three-level system in quantum case. On contrast to the classical memristor in this case the key control parameter is time interval T in equation (10). On Fig. 3 the results for 3-level system interacted with two resonant fields are presented. We add some clarifications in the text:
“Note, that this dependence of y(x) was calculated for tree-level system interacted with two resonant laser fields. It corresponds to the one of the dependencies which were calculated in accordance with (9-10) (please, see Figure 2) It demonstrates principal opportunity to find values of y and x for three-level system in quantum case. It is important that key control parameter is time interval T in equation (10).”
Furthermore, there are some explicit typos, for example “seedback” in abstract. And I have question on Eq. (9). As with Eq.(7) and (8) , $y$ should equals to $1-\gama^2 x$.
Reply: We thank the Reviewer for the comment. It was typo. New version of the manuscript is free from it. We would like to mention, that calculations have been carried out with correct formulae.
Overall, the article is well-written and makes a valuable contribution to the field of quantum computing. However, there may be some areas where the authors could provide more detail or clarification. Without that, I cannot support its publication.
Reply: We thank the Reviewer for the high appreciation of our work.
We thank Reviewer #2 for the useful comments and remarks to the paper. We included the replies to the comments of Reviewer #2 in the revised manuscript, alongside with the replies to Reviewer #1. We hope that, with the above-mentioned modifications of text and figure and the added clarifications, the revised manuscript is now suitable for publication in Entropy.

Round 2
Reviewer 1 Report
The resubmission is still unclear and contains mistakes.
In addition the new text is not very informative and is written in poor english.
I detail below the points to clarify.
1) Below Equation (9), the definition of s is incorrect: It should be s=1-|gamma|^2.
2) It is indicated that s changes according to (10). However it is not clear that (10) can be satisfied from the definition of s and x (7).
Maybe the authors want s to satisfy Eq. (10), i.e. to determine the parameters T, Omega, \tilde Omega such that (10) is satisfied.
3) The protocol below Eq. (11) is unclear. What is the ordering of the laser fields ? I don't see where the laser 1 is involved.
4) Equation (12) is unclear. What does represent <a> ?
I guess that it is |< a | Psi_out >|^2 (?) If this is true, this more precise notation should be used.
5) The numerics:
Figure 3b should be clarified.
It is mentioned that it is y (9) as a function of x (7) for Rabi frequency variation \tilde Omega according to (10-11).
But \tilde Omega is a constant in Eqs. (14-16). How can it satisfy (10-11) ?
6) The value of t_01 should be defined.
The new text in red is written in poor english with many typos.
Author Response
Response Letter
Reviewer #1
The resubmission is still unclear and contains mistakes.
In addition the new text is not very informative and is written in poor english.
Reply: We improved the quality of English and clarified some aspects of the manuscript parts.
I detail below the points to clarify.
1) Below Equation (9), the definition of s is incorrect: It should be s=1-|gamma|^2.
Reply: The value of s in accordance with the equation (1) can be chosen as both s=|gamma|^2 as we do or s=1-|gamma|^2 as the Reviewer #2 suggests. Our definition of s corresponds to (12). If ones want to define s as 1-|gamma|^2 the formula (12) should also be changed.
2) It is indicated that s changes according to (10). However it is not clear that (10) can be satisfied from the definition of s and x (7).
Maybe the authors want s to satisfy Eq. (10), i.e. to determine the parameters T, Omega, \tilde Omega such that (10) is satisfied.
Reply: Formula (10) does not follow from formula (7). As we indicated in the text of the manuscript and in previous responses to the Reviewers comments, the dependence (10) was chosen as close as possible to observable in classical memristors. In this sense, we define |gamma|^2 by formula (10). This can be done, since we ourselves set |gamma|^2 by changing the parameters of the second laser. The only restriction on the form of formula (10) is that it must satisfy the possibilities of the experimental technique.
3) The protocol below Eq. (11) is unclear. What is the ordering of the laser fields ? I don't see where the laser 1 is involved.
Reply: The first laser controls the value of |betta|^2, the second one - |gamma|^2. Both lasers affect the ion simultaneously. Laser 1 is turned on and prepare state x, the laser 2 ensures the implementation of the protocol: by changing its parameters ones can control |gamma|^2 and hence y.
4) Equation (12) is unclear. What does represent <a> ?
I guess that it is |< a | Psi_out >|^2 (?) If this is true, this more precise notation should be used.
Reply: Sure, it is obviously <a>=|< a | Psi_out >|^2. Analogous formula have been used in (7) and (8). It also can be calculated via the density matrix |Psi_out > < Psi_out|.
For more clear understanding, we reformulated the following part of the text.
The following part of the Section 2 was modified in the revised manuscript:
The probability of the ion being in the state $|a\rangle$ in this case equals
P_{a} =|\langle a|\Psi_{out}\rangle|^2=s(0)x.
5) The numerics:
Figure 3b should be clarified.
It is mentioned that it is y (9) as a function of x (7) for Rabi frequency variation \tilde Omega according to (10-11).
But \tilde Omega is a constant in Eqs. (14-16). How can it satisfy (10-11) ?
Reply: The \tilde Omega is a constant for the calculations presented on Figures 3 a and c. As for the results, presented on Fig. 3 b the value of \tilde Omega was changed in accordance with (10-11).
For more clear understanding, we reformulated the following part of the text.
The following part of the Section 3 was modified in the revised manuscript:
“With given values of the field parameters and by changing the value \tilde Omega in accordance with equations (10-11), we calculated y and x. As a result, a hysteresis dependence of y on x was obtained as shown in Figure 3 b.”
6) The value of t_01 should be defined.
Reply: We have added the following definition of both t_01 and t_02 in the revised version of the manuscript.
The following part of the Section 2 was modified in the revised manuscript:
“t_{01} is a center of the pulse of the first laser, t_{02} is a center of the pulse of the second laser”.
We thank Reviewer #2 for the useful comments and remarks to the paper. We included the replies to the comments of Reviewer #2 in the revised manuscript, alongside with the replies to Reviewer #1. We hope that, with the above-mentioned modifications of text and figure and the added clarifications, the revised manuscript is now suitable for publication in Entropy.

Reviewer 2 Report
Thank you for the reply. And at this moment I think I have a better understanding on your article.
Especially for the first question I proposed. It is indeed not crucial to process non-stop measurements. As for this protocol, one should know the ion-trap system very well to process the quantum control, to implement the $s$ function.
For the second question. I have known the motivation that the authors claim the article an Ion-trap protocol. As the authors has said, this could be complicated for Ion trap system, I have a question in my heart, Is this really feasible? I have a suggestion here. If there are not sufficient supporting material for feasibility, can author change the theme of this article to a more modest one. Otherwise, can authors provide more material no matter it is references and reasoning?
This is why I proposed the third question. If the starting point of simulation is an Ion-trap system, for example, its Hamiltonian, dynamical control, and initial state, under such condition, we see a Memristor phenomenon, I will absolutely believe this is an Ion-trap protocol. Otherwise, I regard it as three-level system protocol for Memristor. If the simulation is lacking, as an alternative way, I think the author can make the fourth part abundant, which convincing us that the Ion-trap system could provide such a three-level system evidently.
Overall, the article is improved. After a mild modification, addressing above comments, I can recommend it for publication.
Author Response
Reviewer #2
Thank you for the reply. And at this moment I think I have a better understanding on your article.
Especially for the first question I proposed. It is indeed not crucial to process non-stop measurements. As for this protocol, one should know the ion-trap system very well to process the quantum control, to implement the $s$ function.
For the second question. I have known the motivation that the authors claim the article an Ion-trap protocol. As the authors has said, this could be complicated for Ion trap system, I have a question in my heart, Is this really feasible? I have a suggestion here. If there are not sufficient supporting material for feasibility, can author change the theme of this article to a more modest one. Otherwise, can authors provide more material no matter it is references and reasoning?
Reply: We have focused our interest on ion platform especially because our experimental group is close to realization of this proposal. In the few past years we have developed and demonstrated an ion-based quantum computer operating with up to eight individually addressable 171Yb+ ions [Phys. Rev. A 107, 052612 (2023)] and [“Continuous dynamical decoupling of optical 171Yb+ qudits with radiofrequency fields”; arXiv:2305.06071]. Indeed, the protocols are rather complicated: accurate pulse shaping, frequency control and high-fidelity state reading, besides a high-vacuum system with a rf trap must be under tight control. The missing part in our apparatus is the narrow-line laser (467 nm) which is necessary for the quantum memristor protocol. We are sure that as soon as the laser is available, manipulating with octupole transition will not be difficult as already demonstrated by several groups (e.g. Phys. Rev. Lett. 108, 090801 (2012) and other works from PTB). Note, that at the moment we successfully work with quadrupole transition at 461 nm with the long-living upper state (53 ms).
Relying on our experience, we believe that the memristors on ion-trap system are feasible experimentally. The necessary techniques for the experimental realization of the proposed concept is very similar to ion-base quantum computing and optical atomic clock techniques [26, 27]. Moreover, an ion quantum memristors realization has strong advantage: a capability to build coupled quantum memristors. As a result, one can transfer the memristive state from an ion to an ion producing a neural network. It is much harder to demonstrate this behavior in e.g. NV diamond color centers, where entanglement of two neighbors is really challenging.
We have already mentioned the discussion about a capability to build coupled quantum memristors in the text of the manuscript (please, see section 5 of the manuscript).
The following part of the Section 2 was modified in the revised manuscript:
“Experimental realization of the proposed ion-base quantum memristor concept is rather complicated: accurate pulse shaping, frequency tuning and the high-fidelity state reading, besides a high-vacuum system with a rf trap must be under tight control, as well as manipulating with octupole transition by a narrow-line laser, but it can be done with the technique used for ion-based quantum computing [ 40 ] and atomic clocks. As for the manipulating with octupole transition, it was also already demonstrated [41].
- Aksenov M.A.,et al.. Realizing quantum gates with optically addressable 171Yb+ ion qudits Phys. Rev. A 2023 107 052612.
- Huntemannet N. al. High-Accuracy Optical Clock Based on the Octupole Transition in 171 Yb+ PPhys. Rev. Lett. 2012 108 090801.”
This is why I proposed the third question. If the starting point of simulation is an Ion-trap system, for example, its Hamiltonian, dynamical control, and initial state, under such condition, we see a Memristor phenomenon, I will absolutely believe this is an Ion-trap protocol. Otherwise, I regard it as three-level system protocol for Memristor. If the simulation is lacking, as an alternative way, I think the author can make the fourth part abundant, which convincing us that the Ion-trap system could provide such a three-level system evidently.
Overall, the article is improved. After a mild modification, addressing above comments, I can recommend it for publication.
Reply: In section 4 we suggest one of the possible scheme on Yb ion, parameters of chooses levels are correspond to the necessary demands (in term of temporal decay characteristics to satisfy the inequality (13)) which are originates from the proposed protocol. Equations (14-16) describe population dynamics of ionic levels the proposed three-level system, in particular. The obtained dependences are determined by relative values of parameters, which are quite specific for a real physical system. For example, it is important that the duration of the first laser is significantly longer than the duration of the second one (as in the presented results of numerical studies) which poses certain restrictions.
We thank Reviewer #1 for the useful comments and remarks to the paper. We included the replies to the comments of Reviewer #1 in the revised manuscript, alongside with the replies to Reviewer #2. We hope that, with the above-mentioned modifications of text and figure and the added clarifications, the revised manuscript is now suitable for publication in Entropy.

Round 3
Reviewer 2 Report
Thanks for the authors' reply. I think their answer have answered my questions.